# Fiber and Prebiotic Interventions in Pediatric Inflammatory Bowel Disease: What Role Does the Gut Microbiome Play?

**DOI:** 10.3390/nu12103204

**Published:** 2020-10-20

**Authors:** Genelle R. Healey, Larissa S. Celiberto, Soo Min Lee, Kevan Jacobson

**Affiliations:** Department of Pediatrics, BC Children’s Hospital Research Institute, University of British Columbia, Vancouver, BC V6T 1Z4, Canada; genelle.healey@bcchr.ca (G.R.H.); lceliberto@bcchr.ca (L.S.C.); soomin.lee@bcchr.ca (S.M.L.)

**Keywords:** inflammatory bowel disease, fiber, prebiotic, pediatrics, nutrition, gut microbiome, short-chain fatty acids

## Abstract

The etiology of inflammatory bowel disease (IBD) is complex but is thought to be linked to an intricate interaction between the host’s immune system, resident gut microbiome and environment, i.e., diet. One dietary component that has a major impact on IBD risk and disease management is fiber. Fiber intakes in pediatric IBD patients are suboptimal and often lower than in children without IBD. Fiber also has a significant impact on beneficially shaping gut microbiota composition and functional capacity. The impact is likely to be particularly important in IBD patients, where various studies have demonstrated that an imbalance in the gut microbiome, referred to as dysbiosis, occurs. Microbiome-targeted therapeutics, such as fiber and prebiotics, have the potential to restore the balance in the gut microbiome and enhance host gut health and clinical outcomes. Indeed, studies in adult IBD patients demonstrate that fiber and prebiotics positively alter the microbiome and improve disease course. To date, no studies have been conducted to evaluate the therapeutic potential of fiber and prebiotics in pediatric IBD patients. Consequently, pediatric IBD specific studies that focus on the benefits of fiber and prebiotics on gut microbiome composition and functional capacity and disease outcomes are required.

## 1. Inflammatory Bowel Disease

Inflammatory bowel disease (IBD) is characterized by chronic, relapsing inflammation of the gastrointestinal (GI) tract. There are two main subsets of IBD: Crohn’s disease (CD), which affects any part of the GI tract but is primarily found in the terminal ileum and cecum, and ulcerative colitis (UC), which affects only the colon. The incidence and prevalence of IBD is rising in a concerning fashion, particularly in children [1].

It appears that one of the strongest determinants of IBD risk is a family history of the disease [2], although it is becoming increasingly evident that modifiable environmental factors such as diet, medication use, smoking, hygiene and breastfeeding also play a key role in disease risk [3]. The exact etiology of IBD is unknown, but the current view is that an intolerance to dysregulated gut microbiota (dysbiosis) due to environmental triggers leads to chronic gut inflammation in genetically susceptible individuals [3]. Of the environmental risk factors that have been identified, diet appears to be the most influential. Therefore, a deeper understanding of the impact diet has on the gut microbiome and subsequent development of IBD is warranted.

Presently, there is no known cure for IBD. Therefore, current therapies are directed at controlling symptoms and maintaining disease remission through the use of immunomodulatory medications that unfortunately have side effects and varying success rates. Consequently, novel therapies are in high demand, with alternative and complementary approaches, particularly dietary and gut microbiome-targeted therapeutic approaches, gaining validity and popularity [4]. More well-designed studies are needed to elucidate the true benefit these novel therapies have in treating IBD.

In this narrative review, we discuss the various types of fiber and prebiotics and the benefits they have on human health, the impact low fiber intakes may have on IBD pathogenesis and the suboptimal fiber intakes observed in pediatric IBD patients. We also provide an overview of the studies that have been conducted in adult IBD patients, as presently no relevant pediatric studies have been conducted, describing the impact fiber and prebiotic interventions have on disease outcomes and the gut microbiome.

## 2. Fiber and Prebiotic Types

Dietary fibers (DFs) are commonly known as nondigestible carbohydrates that confer a health benefit to the host. The exact definition of fiber is still a matter of debate and is continuously evolving due to advances in the field and updates to international guidelines for food product definitions [5]. The differences in the chemical structure and physiochemical properties of DF such as water solubility and viscosity, fermentability and specific physiological benefits of each DF make specific definitions a challenge. Moreover, depending on the fiber studied, there is no consensus as to whether the positive health outcomes are restricted to intact fibers within a food matrix or whether the benefits also extend to fibers extracted from plants, i.e., fiber supplements [5].

Most countries’ authorities align with the Codex Alimentarius Commission (CAC) definition established in 2009, where DFs are defined as edible carbohydrate polymers with ten or more monomeric units that are resistant to endogenous digestive enzymes and thus are neither hydrolyzed nor absorbed in the small intestine of humans [6]. The CAC definition provides flexibility to international authorities regarding polymer size; in several countries such as Australia, Brazil, Canada, China, Europe and New Zealand, nondigestible carbohydrates with greater than three monomeric units can be considered DFs [7]. According to the CAC, these edible carbohydrate polymers can be found naturally in whole foods such as fruits, vegetables, legumes and cereals as well as extracted from raw food materials by physical, enzymatic and chemical techniques. Additionally, synthetic carbohydrate polymers can also be consider DFs as long as a physiological benefit is proven [8].

DFs are subdivided into either polysaccharides, such as non-starch polysaccharides (NSP), resistant starch (RS) and resistant oligosaccharides, or into insoluble and soluble forms [8,9]. Soluble fibers are fermented in the colon and thus increase the concentration of beneficial bacterial metabolites such as short-chain fatty acids (SCFAs) [8]. Most soluble NSPs such as guar gum, β-glucans, psyllium and specific pectins are able to absorb water in the intestinal tract, forming a gel structure that improves stool consistency [8,10]. Moreover, the viscosity properties of soluble fibers can delay absorption of glucose and lipids and thus positively impact postprandial metabolism [9]. In contrast, insoluble fibers such as cellulose and hemi-cellulose are poorly fermented by the bacteria present in the colon. Instead, they confer a fecal bulking effect that helps with bowel movements by improving intestinal transit time [8,10]. Soluble and insoluble fibers are found in different proportions among several food categories such as fruits, vegetables, seeds, nuts, legumes and cereals. While common sources of soluble fiber include oatmeal; barley; and legumes such as beans, lentils and peas, insoluble fibers are present mostly in wholegrain cereals as well as in seeds and the skins of fruits and vegetables.

Based on the physiochemical characteristics of DFs it has been suggested that, even as a secondary effect, all fibers positively shift the gut microbiota composition conferring a general health benefit to the host [11]. However, some of these compounds are classified as “prebiotics” as they meet the current International Scientific Association of Probiotics and Prebiotics (ISAPP) definition of “a substrate that is selectively utilized by host microorganisms conferring a health benefit” [12]. While some researchers and clinicians still use the prebiotic and DF definitions interchangeably, it is important to note that while DFs are broadly metabolized and will therefore stimulate growth of an array of gut microorganisms, prebiotics are non-viable substrates that target human and animal-associated microbiota by serving as a nutrient source for specific beneficial microorganisms harbored by the host. Although the resident microorganisms are the predominant utilizers of prebiotics, administered probiotic strains can also benefit from these specific fermentable substrates [12]. Thus, prebiotics have the ability to create a new nutritional niche within the GI tract, providing specific microbes with nutrients allowing them to establish residence within the indigenous ecosystem [13].

Established prebiotics with the most extensively documented health benefits include inulin-type fructans (i.e., inulin, fructo-oligosaccharide (FOS) and oligofructose), galacto-oligosaccharides (GOS) and lactulose [13,14]. Other fermentable carbohydrates that have shown prebiotic potential include RS, *β*-glucans, arabinoxylan oligosaccharides, xylo-oligosaccharides, soy bean oligosaccharides, isomalto-oligosaccharides and pectin [13]. Prebiotics are found naturally in foods such as breads, cereals, onions, garlic and artichokes. Likewise, they are easily incorporated into processed food to increase their fiber content as well as nutritional value. They are also sold in the form of DF supplements [13,15]. Substances such as human milk oligosaccharides and plant polyphenols may also be considered prebiotics due to their ability to positively influence GI health; however, additional studies are required to demonstrate their prebiotic capacity [12].

## 3. Effect of Fiber on Health and Disease

DF intake is notably different across industrialized and unindustrialized countries around the world, with inhabitants of rural communities consuming up to seven times more fiber than typical Western countries due to increased intake of fibrous plants [7]. Western dietary patterns consist of high amounts of processed food rich in sugar and fat and generally devoid of fiber-rich foods such as fruits, vegetables, legumes and wholegrains [16]. Current recommended fiber intakes in Canadian children and adolescents aged 1–3, 4–8 and 9–18 years are 19, 25 [17] and 31 (girls) to 38 (boys) g/day [18], respectively. Current fiber intakes are well below the dietary recommendations, with children 1–3 and 4–8 years consuming only 9.9 and 13.4 g/day, respectively [17]. Meanwhile, adolescent girls and boys are reported to only consume 14 and 16.3–18.2 g of fiber per day [18], respectively, which is less than half of the recommended fiber intake. Low fiber consumption has been indicated as one of the key factors in reducing gut microbiota diversity as well as leading to subsequent increases in chronic non-communicable diseases such as obesity, type 2 diabetes, cancer and cardiovascular disease [19]. Moreover, inadequate fiber intakes are positively associated with pediatric constipation [20] and future health risks including obesity in adulthood [21].

Interestingly, several population studies and meta-analyses have demonstrated a beneficial effect of DFs in the reduction of disease risk, including cardiovascular disease, diabetes, metabolic syndrome, obesity, cancer, diverticular disease, IBD and functional intestinal disorders [22,23,24,25,26,27,28]. The mechanisms of protection in chronic conditions differ depending on the disease as well as the type and dose of the fiber administered; however, shifts in the gut microbiota and the subsequent positive effects of the metabolites they produce (i.e., SCFA) seem to play a major role in the positive outcomes [26]. Prebiotics have also been shown to enhance immune function and insulin sensitivity, which are known to lead to beneficial health effects [29]. Fiber has been reported to help minimize exposure to intestinal carcinogens by diluting fecal content and increasing intestinal transit time [29]. Furthermore, it has been shown that SCFAs produced due to the fermentation of fiber lead to an array of tumor-suppressive and anti-inflammatory effects, with butyrate having particularly compelling effects in cancer [30,31,32] and IBD model studies [33]. Interestingly, some DFs can also aid in weight management and appetite control by improving intestinal transit time, prolonging postprandial and overall satiety and stimulating satiety hormones such as cholecystokinin [34].

## 4. Fiber Intake in Childhood and Adolescence and the Pathogenesis of Inflammatory Bowel Disease

Dietary components, such as fiber, have long been implicated in the pathogenesis of IBD; however, efforts to identify the specific dietary factors responsible have been challenging. Several studies have been conducted to help shed light on the complex interactions that exist between fiber and IBD risk during childhood and adolescence. A study utilizing data collected as part of the Nurses’ Health study II (763,229 person-years of follow-up) was undertaken to determine whether adolescent diets were associated with risk of CD and UC. The effects of dietary patterns, such as a prudent (high intake of fish, fruits and vegetables) or Western (low in fiber-rich foods and high in animal fats and proteins) diet, as well as individual foods or nutrients, were examined. Adolescents with the highest intakes of fish, fruit and vegetables (prudent pattern) had a 53% reduced risk of developing CD. The dietary components with the greatest influence on lowering CD risk were higher fish and fiber intakes. There were, however, no associations between adolescent diet and UC risk [35]. In a case-control study, children newly diagnosed with CD (*n* = 130) were matched to healthy controls (*n* = 202) to evaluate dietary intakes 1 year prior to diagnosis. Similarly, higher intakes of fish, fruits, vegetables and fiber were shown to be protective. Additionally, a higher ratio of omega-3/omega-6 fatty acids was significantly associated with a reduced risk of CD [36]. This group also assessed whether certain dietary patterns were associated with CD risk and found that in girls a dietary pattern characterized by meats, fatty foods and desserts was associated positively with CD. In contrast, a dietary pattern characterized by vegetables, fruits, olive oil, fish, grains and nuts was inversely associated with CD in boys and girls [37]. Notably, a meta-analysis of the association between fiber intake and IBD risk, which included studies with children and adolescents, established that there was a significant inverse relationship between high fiber intake and CD risk. There was, however, only a marginally significant association between fiber and UC risk [38].

The findings from these studies suggest that fiber intake, particularly from fruits and vegetables, protects against the development of CD. To date, the fiber specific studies in UC patients are less convincing. The associations between fiber and IBD risk may, however, be confounded by the fact that individuals consuming higher intakes of fiber, fruits and vegetables are likely concurrently consuming less animal fats and proteins, which are inversely associated with IBD risk. Additional research should focus on demonstrating what fiber types provide protection irrespective of other dietary components and whether fiber interventions can help reduce IBD incidence in high-risk children and adolescents.

The majority of the studies conducted to assess how fiber impacts IBD risk in pediatrics have been conducted retrospectively in children or adolescents that had already been diagnosed with IBD rather than prospectively following children up over time. It is possible that dietary intakes, including fiber consumption, may change leading up to and after IBD diagnosis, making the dietary data collected in these studies less reliable in helping demonstrate whether fiber influences IBD pathogenesis. Large prospective longitudinal studies such as the Genetic Environmental Microbial (GEM) project, which is actively recruiting first degree relatives of CD patients to identify triggers of CD, are desperately needed to help better define how factors such as diet, genetics and the gut microbiome contribute to IBD risk [39].

## 5. Suboptimal Fiber Intakes in Pediatric Inflammatory Bowel Disease Patients

DF intakes in pediatric IBD patients are generally considered to be suboptimal when compared to nutritional guidelines or children and adolescents without IBD. In a cross-sectional study of 57 pediatric CD and 11 pediatric UC patients (a mix of inactive or mild, moderate and severe disease), diet adequacy was assessed by comparing usual intakes with dietary recommendations and healthy control intakes. It was demonstrated that pediatric IBD patients had DF intakes that were significantly lower than the recommended dietary allowance (14.5 ± 9.9 g/day versus 31 ± 6 g/day; *p* < 0.05) and also significantly lower than children and adolescents without IBD (23 ± 1.2 g/day; *p* < 0.05) [40]. In a study conducted in Poland, median total, soluble and insoluble fiber intakes in children with quiescent or mildly active IBD were compared to healthy controls. In contrast, to the above study, children with IBD were shown to have similar fiber intakes to the healthy controls, but both groups had lower than recommended intakes of fiber, with only 24–32% meeting the adequate intake guidelines [41]. Sila and colleagues assessed the fiber intakes of children with newly diagnosed IBD (*n* = 89; 55% with CD) in relation to healthy controls (*n* = 159). Children with UC had significantly lower fiber intakes and children with CD had significantly lower fruit intakes than healthy controls [42]. In a study conducted by Costa and colleagues, it appeared that, regardless of whether a pediatric CD patient had active or inactive disease, fiber intakes were below recommendations, with only around 20% of patients meeting the World Health Organization fiber guidelines [43].

The rationale behind why pediatric IBD patients consume less fiber is likely multifactorial. However, IBD patients are often discouraged from consuming fiber-rich foods, especially during an exacerbation of disease, out of fear of causing further bowel irritation. Some patients have also reported worsening of symptoms due to consumption of high-fiber foods [44]. A survey used to better understand the dietary practices of pediatric IBD patients revealed that 90% of CD and 71% of UC patients had made dietary changes since being diagnosed with IBD. Approximately 8% and 19% of CD and UC patients, respectively, had tried a low-fiber diet, with several high-fiber foods, such as corn and corn products, nuts, bran, tomatoes, raw vegetables, rye, oats and barley being avoided since diagnosis. Of the IBD patients that changed to a low-residue/low-fiber diet, 67% reported a benefit due to the dietary change. The most common symptoms reported to be alleviated were abdominal pain, diarrhea and flatulence [45]. Conversely, in an interventional study, a low-residue diet, which limits high-fiber foods such as wholegrain breads and cereals, nuts, seeds, fruits (raw or dried) and vegetables, led to no differences in symptoms, hospitalization or complications compared to a normal Italian diet high in plant-based foods and DF [46]. Additionally, a diet high in fiber and unrefined carbohydrates was shown to have positive effects in patients with CD and did not increase the risk for intestinal obstruction even in patients with a history of strictures [47]. These studies suggest that avoidance of fiber in IBD patients may not be warranted, especially as there is little evidence of the contrary.

The insufficient fiber intakes observed in children with IBD are important to consider as evidence is beginning to emerge suggesting that inadequate fiber intakes may have detrimental effects on disease course. A study conducted in CD (*n* = 1130) and UC (*n* = 489) patients aimed to demonstrate whether fiber intakes were associated with disease flare over a 6-month period. They observed that patients with CD, but not UC, who continued to consume high-fiber foods were around 40% less likely to have a disease relapse compared to those that avoided these foods. It was outlined in this study that around 30% of patients avoided DF completely, but it was not clear if this was due to medical advice or personal preference. Interestingly, they also identified that individuals with longer disease duration and a history of surgery and hospitalization also had lower fiber intakes [48]. Therefore, recommendations to reduce fiber-containing foods, especially during disease remission, may need to be reassessed. Taken together, it appears that encouraging DF intakes in pediatric IBD patients may need to be considered to help improve clinical outcomes. As very few studies have been conducted to uncover what relationship exists between habitual fiber intakes and disease outcomes, additional research in this area is needed.

## 6. Utilizing Fiber as an Induction or Maintenance Therapy in Pediatric Inflammatory Bowel Disease

At present there is no known cure for IBD. Therapeutic strategies to induce disease remission are, therefore, necessary to alleviate the symptoms associated with disease exacerbation. Conventional medications used to reduce inflammation and prolong disease remission have been associated with detrimental effects on growth and development in pediatric IBD patients [49]. Consequently, novel approaches to relieve IBD-related inflammation are in high demand. Fiber has received recent attention as a candidate treatment option and maintenance therapy in patients with IBD. Research suggests that fiber may help dampen down GI inflammation directly via its effect on the intestinal epithelial cells [50] and indirectly via gut microbiome manipulation of the immune system [51,52] and intestinal barrier function [53,54].

To the best of our knowledge, no pediatric IBD fiber intervention studies have been conducted to date. Consequently, for the purposes of this review, adult IBD fiber intervention studies are summarized to help demonstrate whether there is evidence to suggest that these interventions have potential benefits as treatment strategies in pediatric IBD (Table 1).

### 6.1. Induction Therapy

Clinicians, patients and their families continue to look for more natural diet-based alternative or complementary therapies to help control active disease. The most extensively studied fiber-specific alterative or complementary therapies are inulin and FOS prebiotics. Several studies utilizing different lengths of treatment (3 to 9 weeks) and doses (7.5 g/day to 10 g twice daily) of these prebiotics have been undertaken in patients with active disease [55,56,57,58,59]. The vast majority of these studies showed a significant reduction in disease activity score [56,57,58,59], reduced symptoms [58], enhanced clinical response [59] and higher rates of disease remission compared to the control groups [58,59]. One study did, however, demonstrate that 15 g/day of FOS for 4 weeks led to no significant difference in clinical remission or response when compared to the placebo group [55]. Of note, most of the prebiotic studies undertaken did lead to GI symptoms [55,56,57,59] and at times withdrawal from the study in a subset of patients [55,57,59], but this was not observed in all studies [56,58]. Inulin and FOS are known FODMAPs (fermentable oligosaccharide, disaccharide, monosaccharide and polyol), which in adults with IBD, who have underlying irritable bowel syndrome (IBS), have been shown to worsen GI symptoms [60]. It is, however, unclear whether pediatric IBD patients will experience similar GI symptomology due to inulin and FOS supplementation. This reiterates the need to study the effect of fiber and prebiotic interventions in children with IBD to better understand the clinical benefits and tolerance. Other fiber types have been used during a flare-up of disease, such as whole wheat bran (WWB) [48] and germinated barley foodstuff (GBF) [61]. These fiber interventions, although only used in a small number of studies, have been very well tolerated with no negative impacts reported. Moreover, in the Brotherton and colleagues study, the WWB intervention led to improvements in quality of life (QoL), abdominal pain and diarrhea [62], and for both the WWB and GBF studies, a reduction in disease activity was observed in comparison to control groups [62,63]. At present, strong evidence to either support or discourage the use of fiber interventions in adult or pediatric IBD patients is lacking. Large-scale, robustly designed RCTs are needed to elucidate the true efficacy of fiber interventions in both adult and pediatric IBD patients.

Exclusive enteral nutrition (EEN), a liquid-only diet that allows no food intake, is considered the gold standard treatment option in pediatric CD. EEN is, however, not considered as a first-line therapy in pediatric UC patients [64], but very few studies have been undertaken to determine whether EEN has a beneficial role in the treatment of pediatric UC. A recent study undertaken to provide an extensive nutritional composition analysis of EEN formulas used in the management of pediatric CD identified that only 20% of the formula used contained fiber. Unfortunately, the researchers did not determine whether the addition of fiber to the formula led to more beneficial or detrimental outcomes compared to formula devoid of fiber [65]. In fact, no studies have been conducted to identify whether the addition of fiber to EEN enhances outcomes in pediatric IBD patients. This will be a valuable area of research as, based on the small number of fiber and prebiotic supplementation studies conducted in adult CD and UC patients, the addition of fiber to EEN may further enhance outcomes in pediatric CD as well as improve the efficacy of this nutritional intervention to treat pediatric UC.

### 6.2. Maintenance Therapy

Various fiber intervention studies have been conducted in adult IBD patients with active disease or during remission. For the studies conducted during remission, a combination of both food- and supplement-based fiber interventions have been used. Of the food-based studies undertaken, Chiba and colleagues conducted a study to determine whether a semi-vegetarian diet (SVD), with a fiber content of 32.4 g/day, played a preventative role against disease relapse in CD patients when compared to an omnivore diet (OD). Remission was maintained in 100% and 92% of the SVD group at 1- and 2-years follow-up, respectively, whereas remission was only achieved in 67% and 25% of the OD group at 1- and 2-years follow-up, respectively. Overall cumulative disease relapse rates were significantly lower in the SVD compared to the OD group [66]. The addition of 60 g of oat bran to the diets of remissive UC patients for 3 months also resulted in prevention of disease relapse, but this was also observed in the control group. Unlike the control group, the addition of oat bran did lead to significant improvements in GI symptoms such as abdominal pain and gastroesophageal reflux [67]. Lastly, a randomized cross-over study was conducted in UC patients with remissive or mildly active disease to determine whether a low-fat, high-fiber diet (LFD) enhanced clinical outcomes compared to an improved standard American diet (iSAD; higher quantities of fruits, vegetables and fiber than a typical SAD). The key nutrient difference between these diets was the lower fat content in the LFD; however, the LFD diet was also significantly higher in fiber and lower in sugar than the iSAD. Both diets led to improved QoL, but the LFD also led to a significant reduction in serum amyloid A and a trend towards a decrease in C-reactive protein (CRP) that was not observed in the iSAD group [68]. This suggests that the combination of low fat and high fiber provides additional clinical benefits beyond only increasing fruit and vegetable intakes as recommended in the iSAD group. In all three food-based fiber intervention studies there were high rates of remission maintenance as well as improved clinical outcomes and symptomology [66,67,68]. Additionally, the diets were all well tolerated.

Fiber supplementation has also been used to help prevent disease relapse in IBD patients. A randomized controlled trial (RCT) in UC patients demonstrated that taking a *Plantago ovata* seed (POS) supplement, a fiber supplement known to increase butyrate production, led to similar treatment response rates as mesalamine, a conventional IBD therapy [69]. A GBF supplement was shown to be superior to conventional medications in remissive UC patients, as the GBF group had a significant reduction in CRP and GI symptoms, whereas improvements in these outcomes were not observed in the control group [70]. The use of fiber supplements was, however, associated with some adverse outcomes with one patient in the POS study and three in the GBF study withdrawing due to GI discomfort associated with the supplements. Therefore, it is possible that enhancing fiber intakes via food rather than supplements may be the preferred intervention, as food-based fiber interventions appear to be better tolerated. However, as very few food- and supplement-based studies have been undertaken in remissive IBD patients, more research is required to determine the most efficacious and best-tolerated fiber interventions.

## 7. Fiber and the Gut Microbiome

The gut microbiome is an ecosystem residing within the human GI tract that comprises trillions of microorganisms such as bacteria, archaea, fungi and viruses as well as their respective genes and functions [71]. Diet, particularly fiber, is one of the key factors influencing gut microbiota composition and function [7]. There is a symbiotic relationship that exists between the host and the resident microbiome wherein microorganisms obtain essential nutrients from the host while aiding in DF fermentation and production of beneficial metabolites such as SCFAs [13]. Furthermore, this host–microbiome interaction provides opportunities for targeted dietary modulation of the microbiota since different DFs or prebiotics favor the expansion of certain groups of bacteria [7]. Because DFs promote several metabolic interactions within the gut environment, it is important to consider cross-feeding mechanisms where metabolites produced by fermentation of DF by a particular bacterial species produce secondary substrates for other groups of bacteria to thrive on in the gut. Therefore, cross-feeding reactions can result in several primary and secondary bacterial niches, and this highlights that dietary modulation of the gut microbiome via DF or prebiotics in vivo follows very different mechanistic pathways from in vitro experiments that only utilize single substrates [7].

Dietary patterns directly influence the microbiota composition as well as its effects on the host. A study conducted in healthy American volunteers observed that an animal-based diet rich in meats, eggs and cheeses increased the abundance of microorganisms more tolerant to bile acids (i.e., *Alistipes* spp., *Bilophila* spp. and *Bacteroides* spp.), whereas it decreased the abundance of Firmicutes considered beneficial to the host such as *Roseburia* spp., *Eubacterium rectale* and *Ruminococcus bromii* [72], likely due to a reduced fiber intake. Regarding the necessary time for changes in certain dietary patterns to result in microbiota composition shifts, Smits and colleagues [73] demonstrated that a seasonal reduction in fiber consumption does not appear to have long-lasting effects on the microbiome of the Hazda hunter-gatherer community in Tanzania. However, a long-term reduction in fiber intake, such as seen in Western countries, may lead to extinction of key microbial taxa, as demonstrated in animal studies [74]. Lastly, studies have suggested that a rapid alteration in the gut microbiota composition can occur within 24 h of a dramatic change in macronutrient intake [72]. Additionally, a lower abundance of fiber-degrading bacteria and a reduction in SCFA concentrations have been observed when carbohydrate intakes (including fiber) are reduced [75].

Notably, a systematic review and meta-analysis conducted by So and colleagues [76] explored the impact of DF interventions on gut microbiome composition in healthy adults. The analysis of 64 studies involving 2099 participants concluded that DF interventions resulted in higher abundance of the beneficial bacterial genera *Bifidobacterium* spp. and *Lactobacillus* spp. mainly promoted by fructans and GOS. Moreover, DFs were able to increase fecal butyrate concentrations in comparison with a placebo or low-fiber controls. Interestingly, no significant differences were observed in alpha diversity, abundances of other bacterial species or other SCFAs [76]. Several butyrate-producing bacteria such as *Faecalibacterium prausnitzii*, *Roseburia* spp., *Eubacterium rectale* and *Ruminococcus bromii* have been associated with health benefits and even categorized as next-generation probiotics (NGPs); however, associations between DF consumption and the enhancement of these bacteria in the gut are challenging due to substantial heterogeneity between studies [76].

## 8. Inflammatory Bowel Disease and the Gut Microbiome

The gut microbiome plays a key role in the maintenance of gut homeostasis, nutrient and drug metabolism [77,78], protecting against pathogen colonization [79] and assisting in the regulation of host immunity [80]. However, when perturbation of the symbiotic relationship that exists between a human host and their resident gut microbiota occurs, it may lead to the development of various diseases, including IBD.

A healthy pediatric gut is mainly composed of bacteria that belong to the phyla Firmicutes, Bacteroidetes, Proteobacteria and Actinobacteria [81,82,83]. However, pediatric patients with IBD are observed to have an imbalance in their gut microbiome, referred to as dysbiosis [83]. Several studies have demonstrated that a reduction in microbial diversity and richness is a key feature of both pediatric and adult IBD [81,84,85,86]. Studies have also shown that pediatric patients with IBD exhibit a significantly decreased abundance of Firmicutes and Bacteroidetes while displaying a significant expansion of Proteobacteria [84,87,88,89,90,91]. Within the phylum Actinobacteria, a decrease in the abundance of *Bifidobacterium* spp. has been observed in pediatric IBD patients when compared to healthy controls [86,88]. Indeed, *Bifidobacterium* spp. are known to reduce potential pathobionts and produce anti-inflammatory SCFAs [88]. In comparison, enrichment of Proteobacteria was evidenced to correlate with markers of IBD activity and inflammation, as well as being associated with early relapse [88]. Additionally, members of the Bacteroidetes phylum have been shown to induce the expansion of regulatory T cells and the production of anti-inflammatory cytokines such as IL-10 via the expression of polysaccharide A [91,92]. In summary, the reduction of Firmicutes, Bacteroidetes and *Bifidobacterium* spp., which protect the host against intestinal inflammation, along with the expansion of Proteobacteria, with pro-inflammatory characteristics, are observed in pediatric patients with IBD (Figure 1). Therefore, the interaction between gut microbes and their host is likely to be critical in the pathogenesis of IBD.

Various commensal gut microbiota, including *Bifidobacterium* spp., *Lactobacillus* spp. and members of *Clostridium* cluster IV and XIVa, have the enzymatic capacity to metabolize nutrients that avoid host digestion, such as fiber, and certain amino and fatty acids to produce SCFAs in the colon [93]. Butyrate, which is a SCFA produced via fiber fermentation, is integral in maintaining gut homeostasis [83]. Butyrate has been shown to enhance tight barrier junction protein expression and suppress the production of pro-inflammatory cytokines due to the expansion of regulatory T cells [52,94], thereby protecting the gut from intestinal inflammation. It has been previously reported that patients with IBD display a significant decrease in butyrate concentrations, thus suggesting a strong correlation with the progression of IBD [95,96]. A significant decrease in propionate [95,96], along with an increase in lactate [95] have also been reported in patients with IBD when compared to a healthy cohort (Figure 1). Propionate is a known product of microbial fermentation of lactate by microbes such as *Propionibacterium* in the phylum Actinobacteria and *Clostridium propionicum* in the phylum Firmicutes [97]. Therefore, the decrease in propionate observed in IBD patients may be due to a loss of these metabolizers leading to an accumulation of lactate in the gut. An abundance of lactate in the gut has been previously described to correlate with a higher risk of diarrhea and mucosal inflammation [95].

Dysbiosis appears to be a key feature of IBD, but it is unclear whether the alterations in microbial diversity, taxa abundance and SCFA production are a cause or consequence of intestinal inflammation [86]. Surgical diversion of fecal contents away from inflamed GI segments as well as antibiotic therapy have been shown to be effective in managing IBD, suggesting that the gut microbiome may play more of a causative role in this disease [98,99]. If the gut microbiome is implicated in the pathogenesis and disease course of IBD, then microbiome-based strategies may help reduce the risk of developing IBD in high-risk individuals and provide important therapeutic benefits to adult and pediatric IBD patients.

## 9. The Impact Fiber Has on the Gut Microbiome in Pediatric Inflammatory Bowel Disease

Most IBD-specific exclusion diets, such as EEN, a low-residue diet and the specific-carbohydrate diet, recommend reductions in fiber-containing foods. These exclusion diets have been shown to provide clinical benefits, but they may also have the potential to detrimentally impact the gut microbiome in the long term due to the low fiber content [100]. In fact, EEN has been shown to drive the gut microbiome towards an even more dysbiotic profile characterized by lower bacterial diversity and abundance of bacteria generally considered to be beneficial such as *Faecalibacterium prausnitzii, Bifidobacterium* spp. and *Prevotella* spp. [101]. Utilizing fiber to beneficially modulate the gut microbiome and enhance microbial metabolite production, i.e., butyrate, has the potential to dampen down hyperactive immune responses [52,102] and repair intestinal epithelial barrier dysfunction [103,104,105] to reduce disease activity in pediatric IBD patients.

As mentioned previously, no fiber-specific intervention studies have been undertaken in pediatric IBD patients; therefore, the impact of fiber on the gut microbiome in this patient cohort cannot currently be elucidated. Consequently, for the purposes of this review, we have focused on the adult IBD studies that have investigated the role fiber plays in modulating the gut microbiome (Table 2). The early studies that were undertaken to establish what impact fiber had on the microbiome of IBD patients focused solely on SCFA production. Fernandez-Banares and Hallert and colleagues demonstrated that POS and oat bran interventions, respectively, lead to significant increases in butyrate production, with other SCFAs not being significantly altered [67,69]. Alternatively, a high-dose inulin-type fructan (ITF; 15 g/day) supplement led to a significant increase in total SCFA but only a trend towards an increase in butyrate, with the low-dose ITF (7.5 g/day) having no significant impact on SCFA concentrations. Interestingly, in this study, butyrate was shown to be inversely associated with Mayo scores, a measure of disease activity, in the high-dose group, providing some evidence to suggest that fiber-dependent butyrate production may help treat active disease at least in UC patients [59]. Additionally, a LFD has been shown to alter other microbial metabolites outside of butyrate, with this fiber intervention leading to significant increases in acetate as well as tryptophan [68] (Figure 1).

Fiber intervention studies in adult IBD patients have also been shown to modulate the composition of the gut microbiota, not just the metabolites they produce. The most consistent fiber-specific alteration in gut microbiota composition has been the expansion of bifidobacteria [56,57,59]. Other bacterial taxa that have been shown to be modulated by fiber in IBD patients include *Faecalibacterium prausnitzii*, Bacteroidetes (including *Bacteroides*), Actinobacteria, *Ruminococcus gnavus, Parabacteriodes* and Lachnospiraceae [57,59,68] (Figure 1). One study also showed that a high-fiber diet led to significant shifts in beta diversity from baseline, suggesting that some fiber interventions can have an impact on the entire bacterial community, not just individual bacterial taxa [59]. Conversely, a study providing CD patients with 15 g/day of FOS for 4 weeks led to no significant changes in bifidobacteria or *Faecalibacterium prausnitzii*. Interestingly, this study was one of the only fiber intervention studies in IBD patients that also led to no clinical benefits [55]. It is plausible that the lack of clinical benefits observed in this study was related to the failure of the FOS supplement to modulate the gut microbiome. Other studies have shown that bacterial taxa such as bifidobacteria as well as total bacteria concentrations are positively associated with disease remission and improvements in disease activity. CD patients that entered remission (*n* = 4) after receiving a FOS supplement for 3 weeks were demonstrated to have a significant increase in mucosal total bacteria and bifidobacteria [56]. Additionally, in active CD patients that were given oligofructose-enriched inulin (OF-IN) for 4 weeks, a significant correlation between improvements in disease activity and *Bifidobacterium longum* was observed [57].

Even though clinical benefits have been associated with changes in the gut microbiome, it is very difficult to decipher whether improvements in disease outcomes are truly driven by alterations in the gut microbiome. Studies that utilize more sophisticated sequencing and bioinformatic techniques in combination with robust animal experiments may help provide additional clarification on the causal role the gut microbiome plays in IBD. Lastly, no studies have been undertaken to determine the impact fiber has on the mycobiome (fungi) or virome (virus) of IBD patients and whether changes in these microorganisms are associated with enhanced clinical outcomes. Utilization of microbiome analysis techniques such as shotgun metagenomic sequencing will provide in-depth data on the influence fiber has on the composition and functional capacity of gut-associated bacteria, viruses and fungi.

In summary, the results generated from the small number of studies that have investigated the impact fiber has on the gut microbiome of IBD patients suggest that fiber could be utilized as a microbiome-targeted therapeutic strategy to reduce dysbiosis and enhance clinical outcomes in IBD patients. There are still large gaps in our knowledge on the impact fiber has on the microbiome of children with IBD, what fiber types and doses are the most efficacious and whether fiber interventions can be utilized to beneficially modulate microorganisms other than bacteria, such as fungi and viruses.

## 10. Conclusions

To the best of our knowledge, all fiber and prebiotic intervention studies in IBD patients have been conducted in adults. Therefore, a major knowledge gap relating to the efficacy of these interventions in a pediatric patient cohort exists. The majority of studies undertaken in adult IBD patients do suggest that fiber and prebiotic interventions are able to relieve GI symptoms, reduce disease activity, increase the length of time in remission and improve QoL. One study was, however, unable to demonstrate any positive outcomes associated with a FOS supplement in CD patients. Interestingly, this was one of the only studies that was unable to show any significant changes in the gut microbiome due to a fiber or prebiotic intervention. Was the lack of clinical benefits observed in this study related to the failure of the FOS supplement to modulate the gut microbiome? This is a question that certainly warrants further investigation. Even though there is some heterogeneity in results between studies in adult IBD patients, further investigation in a pediatric IBD cohort is definitely warranted.

Both pediatric and adult IBD patients appear to have dysbiotic gut microbiome profiles when compared to healthy individuals. Therefore, interventions known to beneficially modulate the gut microbiome, i.e., fiber and prebiotics, may help to restore gut homeostasis and positively impact IBD outcomes. It is believed that fiber can dampen down the immune system and strengthen intestinal epithelial barrier function via modulation of the gut microbiome to produce health-promoting metabolites such as SCFAs. Interestingly, several studies have shown that bifidobacteria and the SCFA, butyrate, are inversely associated with markers of disease activity, suggesting certain bacterial taxa and their metabolites may play a protective role in IBD.

Unfortunately, some fiber and prebiotic supplements were not well tolerated by the IBD patients included in the studies outlined in this review. This may be due to some IBD patients having concurrent IBS of which various fiber types are known to exacerbate GI symptoms. It is unclear whether pediatric IBD patients would experience similar IBS-related symptoms due to fiber and prebiotic supplementation. This further establishes the need to undertake well-designed fiber and prebiotic intervention studies in the pediatric IBD population.

At present, there is an absence of conclusive evidence to either support or discourage the use of fiber- and prebiotic-containing foods or supplements by adult or pediatric IBD patients, either during remission or active disease. Preliminary results in adult IBD patients provide some evidence to suggest that these dietary strategies may enhance outcomes in IBD patients. The true efficacy of fiber and prebiotic interventions does, however, need to be elucidated before these potential therapeutics can be widely recommended to IBD patients. Therefore, large-scale, robustly designed fiber- and prebiotic-specific RCTs are in high demand, particularly in the pediatric population where to date no relevant studies have been conducted.

## Figures and Tables

**Figure 1 nutrients-12-03204-f001:**
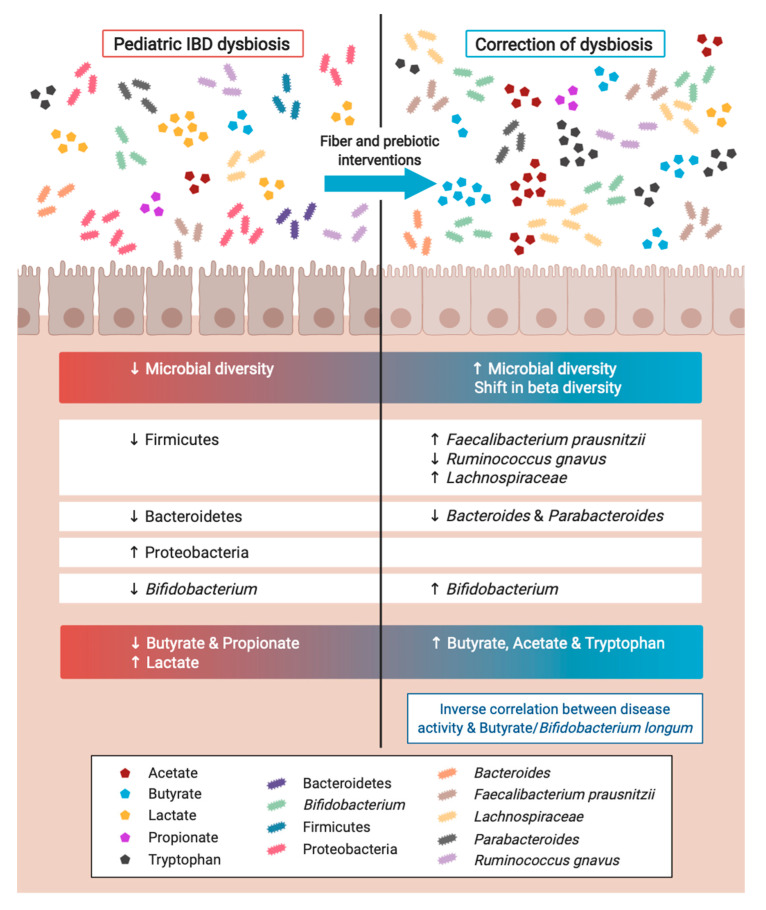
Pediatric IBD patients appear to have a dysbiotic gut microbiome profile compared to healthy controls. Based on the results generated from adult IBD studies, there is potential for fiber and prebiotic interventions to correct the dysbiosis observed in pediatric IBD patients. Created with BioRender.com. IBD: inflammatory bowel disease.

**Table 1 nutrients-12-03204-t001:** Food- and supplement-based fiber and prebiotic intervention studies in adults with IBD in remission or during active disease.

Intervention	Duration	Study Type	Disease	Participants	Tolerance	Key Clinical Outcomes	Reference
Remission							
SVD vs. OD. DF content SVD—32.4 g/day	Up to 2 years	Prospective intervention study	CD	*n* = 16 on SVD and *n* = 6 on OD (median age 26.5; range 19–77 years)	No untoward effects with SVD diet	100% remission maintenance on SVD diet after 1 year and 92% after 2 years vs. 67% and 25%, respectively, on the OD. Cumulative relapse rates significantly lower in SVD vs. OD after 2 years.	[66]
(1) POS 10 g bd or (2) MES 500 mg tds or (3) both	1 year	Randomized controlled trial	UC	*n* = 35 on POS, *n* = 37 on MES and *n* = 30 on both (median age 39.7–46 years)	1 in POS and 2 in both withdrew with constipation and/or flatulence	No difference in probability of maintaining remission at 1 year between groups—treatment failure rates were 40%, 35% and 30% for the POS, MES and both groups, respectively.	[69]
60 g oat bran (20 g/day DF) daily	3 months	Prospective intervention study	UC	*n* = 19 oat bran and *n* = 10 controls (mean age 43.5; range 20–77 years)	Well tolerated	No signs of disease relapse—oat bran or control groups. Significant improvement in GI symptoms (abdominal pain and reflux) in oat bran group. Controls had an increase in reflux.	[67]
10 g tds GBF	2 months	Prospective intervention study	UC	*n* = 20 GBF and *n* = 21 controls (mean age 33.04–33.9 years)	3 patients withdrew due to GI discomfort	Significant reduction in CRP in GBF group. Significant improvement in symptoms (abdominal pain and cramping) in GPF group. No significant improvements in CRP or symptoms in control group.	[70]
LFD or iSAD	4 weeks with 2-week washout	Randomized cross over study	UC—remissive and active disease	*n* = 17 (median age 41.7 years)	Both diets were well tolerated	All patients remained in remission during study. Both diets improved QoL. Serum amyloid A significantly decreased in LFD but not iSAD group. Trend towards a decrease in CRP in LFD group.	[68]
Active disease							
15 g/day FOS	4 weeks	Randomized double-blinded placebo-controlled trial	CD	*n* = 54 FOS and *n* = 49 placebo (mean age 39.5 years)	10 in FOS and 3 in placebo withdrew—worsening symptoms	No significant difference in clinical remission (CDAI ≤150) or response (fall in CDAI by ≥70) between FOS and placebo. Increased DC staining of IL-10 in FOS group.	[55]
15 g/day FOS	3 weeks	Prospective intervention study	CD	*n* = 10 FOS (mean age 40; range 29–46 years)	Significant increase in gut rumbling and flatulence severity. No withdrawals	Significant reduction in disease activity (HBI). Significant increase in IL-10 positive DC and DC’s expressing TLR2 and TLR4.	[56]
10 g bd of OF-IN	4 weeks	Randomized double-blinded placebo-controlled trial	CD—remissive and active disease	*n* = 34 OF-IN and *n* = 33 placebo (age not specified)	High withdrawal due to side effects—*n* = 10 in OF-IN and *n* = 3 in placebo	8 patients with active CD in OF-IN group had significant reduction in disease activity (HBI).	[57]
WWB (1/2 cup/day) with reduced refined CHO	4 weeks	Prospective interventional study	CD	*n* = 4 WWB and *n* = 3 control (age not specified)	No negative effects reported	WWB had greater improvement on QoL. Reduction in disease activity (HBI) over time in WWB group. Improvement in abdominal pain and reduction in diarrhea in WWB but not control group.	[62]
MES +/− 12 g/day ITF	2 weeks	Randomized double-blinded placebo-controlled trial	UC	*n* = 9 MES and *n* = 10 MES + ITF (median age 36.5; range 30–44 years)	Well tolerated	Significant reduction in dyspeptic symptoms in ITF group. Significant reduction in fecal calprotectin in the ITF group. All ITF participants went into clinical remission whereas 2 participants in the MES showed continued disease activity.	[58]
7.5 g/day or 15 g/day of ITF	9 weeks	Randomized controlled trial	UC	*n* = 12 7.5 g/day ITF and *n* = 13 15 g/day ITF (mean age 37.3; range 18–65 years)	1 from each group withdrew—worsening symptoms. 6 in 15 g/day and 1 in 7.5 g/day group reported flatulence and bloating—transient and reduced over study	Clinical response (change in Mayo score) was shown in 77% and 33% of the 15 g/day and 7.5 g/day groups, respectively. 8 vs. 2 patients went into clinical remission in the 15 g/day vs. 7.5 g/day groups, respectively. Significant reduction in fecal calprotectin in the 15 g/day group.	[59]
GBF 20–30 g/day	4 weeks	Randomized controlled trial	UC	*n* = 11 GBF and *n* = 7 control (median age 37 years)	No side effects reported	Significant decrease in clinical activity index score in GBF group compared to controls.	[63]

bd—twice daily, CD—Chron’s disease, CDAI—Crohn’s disease activity index, CRP—c-reactive protein, DC—dendritic cell, DF—dietary fiber, FOS—fructo-oligosaccharide, GBF—germinated barley foodstuffs, GI—gastrointestinal, HBI—Harvey Bradshaw index, IBD—inflammatory bowel disease, IL—interleukin, iSAD—improved standard American diet, ITF—inulin-type fructan, LFD—low-fat, high-fiber diet, MES—mesalamine, OD—omnivore diet, OF-IN—oligofructose-enriched inulin, POS—*Plantago ovata* seeds, QoL—quality of life, SVD—semi-vegetarian diet, tds—three times daily, UC—ulcerative colitis, WWB—whole wheat bran.

**Table 2 nutrients-12-03204-t002:** Fiber and prebiotic intervention studies in adult IBD and their impact on the gut microbiome.

Intervention	Duration	Study Type	Disease	Participants	Analysis Methodology	Key Microbiome Outcomes	Reference
Remission							
(1) POS 10 g bd or (2) MES 500 mg tds or (3) both	1 year	Randomized controlled trial	UC	*n* = 7 POS	GC—stool taken from rectum using rectoscopy at baseline and post intervention	Significant increase in butyrate after POS (6.1 to 9.2 μmol/g). Trend towards an increase in acetate.	[69]
60 g oat bran (20 g/day DF) daily	3 months	Prospective intervention study	UC	*n* = 19 oat bran	GC—stool collected every 4 weeks	36% increase in butyrate after 4 weeks on oat bran. No significant differences in other SCFA.	[67]
LFD or iSAD	4 weeks with 2-week washout	Randomized cross-over study	UC—remissive and active disease	*n* = 17	16S rRNA sequencing and LC-MS—stool collected at baseline and post intervention	Trend towards an increase in Faith’s alpha diversity after LFD. Significant shift in beta diversity from baseline in LFD group but not iSAD. LFD led to a significant increase in Bacteroidetes and significant decrease in Actinobacteria. *Faecalibacterium prausnitzii* was increased in the LFD group. Acetate increased after the LFD and iSAD. Tryptophan decreased on the iSAD and increased on the LFD.	[68]
Active disease							
15 g/day FOS	4 weeks	Randomized double-blinded placebo-controlled trial	CD	*n* = 54 FOS and *n* = 49 placebo	FISH—fresh stool samples at baseline and post intervention	No significant differences in bifidobacteria or *Faecalibacterium prausnitzii.*	[55]
15 g/day FOS	3 weeks	Prospective intervention study	CD	*n* = 8 FOS	FISH—stool and mucosal biopsy samples at baseline and post intervention	Significant increase in stool but not mucosal bifidobacteria. No significant changes in total bacteria, *Clostridium coccoides*, *Eubacterium rectale* or *Bacteroides*. Those who entered disease remission (*n* = 4) had an increase of mucosal total bacteria and bifidobacteria.	[56]
10 g bd of OF-IN	4 weeks	Randomized double-blinded placebo-controlled trial	CD—remissive and active disease	*n* = 34 OF-IN and *n* = 33 in placebo	Real-time PCR—stool sample	OF-IN led to a significant decrease in *Ruminococcus gnavus* and increase in *Bifidobacterium longum*. No significant change in *F. prausnitzii*. Significant positive correlation between improved disease activity (in active CD) and *B. longum* in OF-IN group.	[57]
7.5 g/day or 15 g/day of ITF	9 weeks	Randomized controlled trial	UC	*n* = 12 7.5 g/day ITF and *n* = 13 15 g/day ITF	Roche 454 sequencing and GC—stool and mucosal biopsy samples	No significant clustering on PCA between treatment groups. Significant increase in stool Lachnospiraceae and Bifidobacteriaceae in high-dose ITF group. Significant reduction in mucosal *Bacteroides* and *Parabacteroides* in high-dose group. Significant increase in total SCFA and a trend towards an increase in butyrate in high-dose group. Butyrate was significantly inversely associated with Mayo score.	[59]

CD—Chron’s disease, DF—dietary fiber, FISH—fluorescent in situ hybridization, FOS—fructo-oligosaccharide, GC—gas chromatography, IBD—inflammatory bowel disease, iSAD—improved standard American diet, LC-MS—liquid chromatography–mass spectrometry, LFD—low-fat, high-fiber diet, MES—mesalamine, OF-IN—oligofructose-enriched inulin, PCA—principal component analysis, PCR—polymerase chain reaction, POS—*Plantago ovata* seeds, rRNA—ribosomal ribonucleic acid SCFA—short-chain fatty acids, tds—three times daily, UC—ulcerative colitis.

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
