# Peer review of "Fiber and Prebiotic Interventions in Pediatric Inflammatory Bowel Disease: What Role Does the Gut Microbiome Play?"

_nutrients, 2020, doi:10.3390/nu12103204_

Round 1

Reviewer 1 Report

This review presents interesting data pertaining to the therapeutic potential of fiber and prebiotics in pediatric IBD patients.  Overall, this is a clear, concise, well-written and with updated references. The review is relevant and sufficient information about novel findings is presented for readers.  However, there are some errors of “Tense and Grammar” throughout the manuscript. Therefore, an editing is needed in order to fix the ENGLISH language.

Reviewer 2 Report

This paper, entitled “Fiber and Prebiotic Interventions in Pediatric Inflammatory Bowel Disease: What Role Does the Gut Microbiome Play?” evaluates the potential of fiber and prebiotics in pediatric IBD patients. The study is very interesting and provides useful information on the role of fiber in the pathogenesis of IBD and their potential role in the therapy of pediatric patients.

The matter of the revision is highly relevant and the paper is presented very clearly.

However I have some concerns:

  1. The authors didn’t indicate whether they conducted a systematic review or a personal review of the literature. Anyhow, they should indicate the key-words/mesh-terms and the databases used for the literature search. Please include a method section that indicate keywords and type of search (including the used databases: i.e.: pubmed, EMBASE, Web of Science, Mendeley, etc. ?).
  2. In the case of a systematic review, the PRISMA flowchart indicating the process of selection of the studies should be included.
  3. Nutrition is used as therapeutic strategy in pediatric patients with Crohn’s disease. In particular elemental diet can be as effective as steroids in the induction of remission (Narula N, et al. Cochrane Database Syst Rev. 2018 Apr 1;4(4):CD000542.). The explanation for the therapeutic effect is thought to be the reduction of the antigenic load. What can be said about the effect of the fibers and prebiotics?
  4. Paragraph 6. Studies on the induction and maintenance of remission are mixed together. Two distinct subparagraph should be distinguished: (1) induction of remission (active disease), and (2) maintenance of remission. Regarding the studies on active disease, further to measurement of clinical activity, information about C-reactive protein, fecal calprotectin, endoscopic and histologic activity should be underlined (if available). Furthermore, information on improvement of growth failure should be added (if available). Please add it.

Minor point: I think that the last sentence of the abstract should be rephrased: “studies focusing on the impact fiber and prebiotics have on the gut….”.
